# Common Traditions, Practices, and Beliefs Related to Safe Motherhood and Newborn Health in Morocco

**DOI:** 10.3390/healthcare11050769

**Published:** 2023-03-06

**Authors:** Chaimae Moujahid, Jack E. Turman, Loubna Amahdar

**Affiliations:** 1Laboratory of Health Sciences and Technologies, Higher Institute of Health Sciences, Hassan First University of Settat, Settat 26000, Morocco; 2Department of Social and Behavioral Sciences, Richard M. Fairbanks School of Public Health, Indiana University, Indianapolis, IN 46202, USA; 3Department of Pediatrics, School of Medicine, Indiana University, Indianapolis, IN 46202, USA

**Keywords:** maternal health, postpartum practices, healer, faith, witchcraft, Morocco

## Abstract

The cultural context influences women’s antenatal care and postpartum knowledge. This study aims to determine the traditional practices related to maternal health in Morocco. We conducted in-depth qualitative interviews with 37 women from three different Moroccan regions on the first postpartum day. We used thematic content to analyze data, and an a priori coding framework was created utilizing the pertinent literature. Beliefs regarding pregnancy and postpartum positively affect maternal health, such as family support, prolonged rest for health recovery, and specific dietary precautions according to the mode of delivery of the new mother. However, some practices may negatively affect maternal health, such as cold postpartum treatment through traditional medicine and not seeking prenatal care after the first pregnancy experience. Such practices include painting newborns with henna, using kohl and oil to hasten the umbilical cord’s descent, and producing solutions based on chicken throat to cure respiratory ailments in newborns that might harm their health.

## 1. Introduction

Morocco has been able to progress in reducing maternal mortality. According to the World Health Organization, this progress reached 61.8% in Morocco between 1990 and 2015 [1]. Achieving the SDG target of a global MMR below 70 requires reducing the global MMR by an average of 7.5% annually between 2016 and 2030 [1]. This progress requires more than three times the 2.3% annual rate of reduction observed globally between 1990 and 2015. Wide disparities in maternal health between urban and rural areas must be addressed to meet these goals [2].

Most maternal and newborns deaths are caused by sociocultural variables, such as cultural and religious influences, and other social factors that impact personal choices. Every culture has certain enduring customs that represent societal values and have been passed down from generation to generation [3,4,5]. Women are influenced by cultural and traditional norms of their ethnic origin regardless of whether they reside in an urban or rural location. This influences how they adjust to the postpartum time, and how they see providing postpartum maternal and infant care [6,7]. 

We grounded our work on a theoretical framework based on an ontological starting point and understanding of the lifeworld of the women as created from their behaviors, attitudes, practices, and perceptions and community and gender norms [8,9]. To improve Moroccan maternal and child health outcomes, it is imperative to investigate how these women view their prenatal and postpartum situations [10].

The interaction between study participants and the researcher serves as the epistemological starting point for knowledge creation about their lived prenatal and postpartum experiences. It is possible that lifeworld knowledge is so ingrained in the culture that community members are unaware of its consequences for women’s and children’s health. Our study helps in revealing this for populations in Morocco [11] to improve outcomes for women, children, and families, and their healthcare team.

To identify harmful traditional practices and provide guidance on counseling women during antenatal and postnatal education to improve maternal and infant health, nurses and midwives would benefit from understanding traditional practices and beliefs related to maternal and infant care. There is, however, little research on Moroccan customs associated with the postpartum period [12,13]. In this study, we sought to understand the traditional practices of women regarding maternal health during pregnancy and after delivery.

## 2. Materials and Methods

### 2.1. Setting 

This study was performed in hospitals from three provinces between March and September 2021. The ethics committee of the Faculty of Medicine and Pharmacy of Rabat, Morocco approved the study (reference CERB: 45/21).

### 2.2. Participants 

Participants were 37 postpartum married women from the Essaouira (*n* = 13), Settat (*n* = 13) and Salé (*n* = 11) provinces. Each selected woman had at least one child and was accepted to participate in the study. The participants were 17–37 years old, and they had had at least one experience with childbirth. Two were employees, ten were illiterate, and four had had a miscarriage experience. All women lived at home with their husbands, and most of them with the husband’s family.

### 2.3. Materials 

The materials consisted of in-depth interviews with participants during their postpartum stays, written documents (personal notes and comments), and observations along the study were adopted. The interviewer used thematic guides covering the following aspects: the new mother’s diet, rest, and activity, the newborn’s diet, beliefs, and practices related to the health of the mother–newborn couple, and the perception of their benefits and drawbacks. We conducted the interviews in an Arabic dialect, and each interview lasted between 40 and 90 min. All interviews were recorded with the permission of the interviewees to be transcribed and analyzed afterward. 

### 2.4. Data Analysis

A digital voice recorder and a smartphone were used to record the qualitative data. Data analyses were performed concurrently with data collection with the goal of concluding the study when new obtained data no longer provided additional insights. The method involved reading and rereading the interviews and focusing on the identification of themes linked to intrapartum and postpartum continuum of care beliefs and practices at each research setting. Data collected in the local language were first translated into French and then to English by two researchers; the interviewer and another person compared the translations for consistency. Using computer-assisted qualitative data analysis software (CAQDAS) MAXQDA 2020, we coded the verbatim transcripts of interviews and field notes to identify emerging themes. 

We classified and organized data using a conceptual framework on the basis of developing themes, concepts, and categories [14]. The writers and investigators reviewed and reread the transcripts individually to discover emergent patterns, and a framework was constructed on the basis of these themes. To build explanations and uncover relationships, data segments were used for each item to characterize comparable and different beliefs and find associations.

From preliminary reading of the transcripts, a coding scheme for the key themes and subthemes was created. The coding scheme specified the numerous codes used during coding, their definitions, when to use and when not to use such a code, and examples of statements to consider while coding into certain codes. These codes were converted into software nodes. Following that, all transcripts were read line by line within the MAXQDA 2020 program, and important elements of the respondents’ utterances were coded into existing and new nodes. The numerous codes were classified into themes on the basis of codes to reflect the content of the material obtained and memos written on certain pertinent topics.

## 3. Results

The following section illustrates the significance of themes from the interviews that presented the living experiences and understanding of 37 Moroccan women’s cultural and traditional health beliefs related to pregnancy, childbirth, and newborn health. Three major themes emerged from the analysis: cultural practices related to pregnancy, cultural practices related to safe motherhood, and cultural practices described to newborn health. 

### 3.1. Cultural Practices Related to Pregnancy

If the first pregnancy is safe, the later pregnancies do not need to be followed up. In contrast, a sex change between two successive pregnancies negatively affects the woman’s health, and she would suffer in the subsequent pregnancy: “*… This time I had pain in my belly because I got pregnant with a boy after a girl. It says that is because of a change of sex of the baby between two successive pregnancies…*”—Khadija.

Protecting pregnant women from nasty smells, intense emotions, and other unpleasant things is essential. They should eat nourishing food; the husband should bring back the food for which the pregnant woman asks. Otherwise, stains would appear on the newborn’s body. The neighbors and the family should help the pregnant woman with the housework, especially in the first and last trimesters.

Muscular uterine contractions are a sign of childbirth, weak contractions are a sign of difficulty in birth, with the mother-in-law giving advice. “*Before the delivery, I smoked myself with the peganum “lharma”, my mother-in-law insisted on thwarting contraction to facilitate and accelerate the delivery.*”—Zahra.

### 3.2. Cultural Practices Described to Safe Motherhood

#### 3.2.1. Witchcraft and Sorcery—“Chaawada”

Women believe in the practice of witchcraft that negatively affects the newborn and new mother’s health; participants confirmed that a course related to witchcraft exists.

Chaawada witchcraft is a practice from which the mother–newborn couple has not been spared, so the woman hides the newborn from the hospital to the house from visitors, except for the closest family who can see the baby.

“*We must be careful from the puerperal blood “dem nfass”, the woman must do her preventive measures so that only people whom trust take care of its sanitary napkins… We risk people hurting us and doing sorcery by using the napkin of the new mother delivered. What can they do with the blood of a new mother? To separate the woman from her husband and not to have more children…*” —Latifa

#### 3.2.2. Bad Eyesight

Some women find that the Quran is an effective way to protect the new mother “nafssa” and the newborn. Without the need to practice customs to fight against the evil eye, others mentioned that the mother or the mother-in-law had the right to practice her traditions to protect the newborn and the new mother.

Fumigations based on alum and peganum “chebba” and “harmal” are necessary during the seven days of postpartum each day before sunset. “*… We put 101 cynoge in the newborn’s right hand, and they fumigate it with alum and peganum CHEBBA and HARMAL, which protects the newborn against the evil eye.*”—Ghizlan.

#### 3.2.3. The 40 Days of Postpartum—“Trabaina”

In the Moroccan context, the new mother should not leave her home before the 40th day, “trabaina”. The newborn’s vaccine comes initially in the first 40 days of postpartum.

Someone in the family oversees bringing the newborn back to the medical facility for the vaccine. 

“*… the newly delivered woman should not go out before the 40 days of postpartum it is for its health so that it does not have the bad eye, she should not speak too much… she should not walk nor make the household… she should just rest and behave sick.*”—Khdija

The postpartum consultation is no longer necessary when she feels well.

The first 40 days of postpartum, “trabaina”, is a vulnerable period where the new mother is exposed to any risk of complications “one foot in the ground and the other in the grave”. Once she exceeds the 40 days, there is no more risk, and the woman returns to everyday life. She can go out and travel with her baby and resume household chores: “*… and I return to the housework after I finish 40 days of postpartum.*”—Samira.

#### 3.2.4. Postpartum Hemorrhage—“Lfayda”

Postpartum hemorrhage, “lfayda”, is a sign of danger. It is a complication that requires care in a health facility; women say that the bleeding kills the nafssa if she is not hospitalized, and the risk of death is significant. Hospital delivery is preferable to home delivery; the traditional midwife can no longer manage the situation in case of postpartum hemorrhage.

#### 3.2.5. The “Nfass”, "Bard Nfass”, or “Tajoughit” Cold

The cold of “nfass” or “bard nfass” negatively affects the health of the new mother, the “nafssa”. If she is exposed to the cold during the postpartum period, does not warm her body, and does not tighten her head scarf, whatever the season, she loses consciousness and becomes insane. It is irreversible, and “bard nfass” can kill the woman. Otherwise, it causes side effects in the long term, such as chronic migraine in its mildest forms.

There are two types of “bard nfass”, the one attacks the head and the second affects the uterus—“iwalda”; warming via preparations based on medicinal plants is an effective way to treat it.

“*The woman must pay attention and take her measures of protection against the cold… if she had a cold after having sweated, she will have disabling complications going until death, ‘kaytnfakh liha rasha o tkdar tmout’.*” —Saadia

“*The cold of postpartum ‘tajoughit’ or ‘tajdayt’ or ‘tajjayt’ is hazardous for the health of the new mother. It is not treated in the hospital by medicines, it requires a tradition such as Timija and Tasrghint. The woman must smoke these plants to treat the cold that attacks her after the childbirth until the tears flow from her eyes. Women must tighten her head very well with a napkin, she doesn’t expose to the light.*” —Habiba

#### 3.2.6. Nutrition of “Nafssa”

Women say that “rfissa” is the principal meal of the new mother in the Moroccan setting. The soup of chickens in the open air, spices “lmsakhen”, and raw and cooked eggs, milk, and peppergrass seeds serve to warm the new mother’s body and the production of breastmilk.

Only women who have given birth via vaginal delivery without episiotomy have the right to eat these dishes to warm the body to avoid the risk of suture disunion and bleeding. 

“Labrik” is an herbal broth whose essential plant is Alpinia galanga “khounjlan” and Madder “lfoua,” which helps in eliminating blood debris from the uterine cavity after birth. Meals composed of grilled liver and minced meat help restore blood lost during delivery.

“*I must eat “rfissa” to regain her physical well-being, but now that I have had a cesarean section, I should not eat the hot foods that include “lmsakhen” since it promotes bleeding… When I gave birth to my first child, they gave me the same as other women; I ate hot meals.*”—Ghizlan

“*… so, they prepared me a mist of medicinal plants that warms the womb “lwalda” it is a “lbrik,” it is a broth-based on khounjlan and el foua “this broth increases the blood fluidity … and elfoua treats anemia and jaundice… The khounjlan sterilizes the uterine body of the woman.*”—Malika

#### 3.2.7. New Mother’s Bath: The Day of Tightening—“Nhar L Hzam”

Therapeutic herbs are used to treat postbirth pain. Hot baths are not recommended for women delivered with a Cesarean section or episiotomy. Nafssa believes that, after birth, the bones of the woman’s body move and hot bath with body massage help to returns the bones to their original position. “*… The Tayaba goes up on the back of Nafssa, makes the massage, and tightens the body from feet to head so that the bones can return to their place, because the woman’s bones shift after the delivery.*”—Ilham

### 3.3. Cultural Practices Related to Newborn Health

#### 3.3.1. New Clothes Are Bad Luck

The first clothes that the newborn wears should be old and have been used by a living baby, as there is the belief that new clothes are a stroke of bad luck, and the mother should not buy new clothes until she delivers her newborn alive; they do not advance joy anymore. 

#### 3.3.2. Sorra

The participants say that, if someone enters the couple’s mother–baby room and wears anything tight, the newborn will have “sorra”, a deformation and swelling of the child’s head with an increased cranial perimeter… Women bring their babies to the fqih (religious healer) or rebagha (healer woman) three times and treat sorra with inflamed sticks of white marrube and mariout or henna. “*… When my children have the sorra, we take them to the fkih who has the baraka to do the necessary and burn them with something. It is my mother-in-law who takes care of this task.*”—Halima.

#### 3.3.3. LAGHROUR

An oily preparation based on argan oil, butter, olive oil, seven types of medicinal plants, and hull throat is essential for every birth; a spoonful of this preparation helps in clearing lung secretions and lubricating the throat of the newborn.

“*“Laghrour” is prepared either with argan oil, butter “zebda l beldia”, or olive oil it is a mixture of plants such as lavender, garlic, thyme, Marrubium… with the throat of the chicken, it is a beneficial treatment for respiratory crises, influenza, nasal obstruction, for respiratory attacks, flu… it is given to the baby in the 3rd day after birth*”—Latifa

#### 3.3.4. Umbilical Care

Umbilical care with oil and henna powder or with kohl accelerates the dryness and then fall of the umbilical cord when medical treatment is not effective.

#### 3.3.5. Ritual Practices

On the seventh postpartum day, the mother-in-law submerges the newborn’s body in henna. To eliminate the skin’s desquamation and prevent neonatal icterus, the baby takes a bath only after one day of this ritual practice. Participants indicated that honey is equivalent to all necessary vaccination doses.

“*… at the 7th day, the whole body of the newborn is taken out in the henna, and then we dry it with a towel without washing it. We do this practice to eliminate his skin’s dermal cortex and prevent jaundice. My mother and mother-in-law prepare a collyrium based on saffron at home to cure the baby’s eye infections…*” —Halima

## 4. Discussion

### 4.1. Cultural Practices Related to Pregnancy

Pregnancy is a privileged event in Moroccan society; the pregnant woman receives help during pregnancy from those around her and eats nourishing food. Eating protein-rich foods, family care, and support for Moroccan women throughout pregnancy and after birth are all likely beneficial from health and social standpoints. Similar findings were seen in other cultures in Zambia and China [3,6].

Morocco has seen a decline in the utilization of health services by women during pregnancy, according to an ENPSF study [2,15]. Connecting pregnancy pain with infant sex change is one of the beliefs that might have a detrimental impact on maternal health. Health professionals must emphasize these sociocultural behaviors during health education and devise focused interventions.

In the Moroccan context, heat-up contraction and fumigations stimulate contractions and strengthen them; this was also reported by [12,13,16] to hasten the active phase before going to hospital. This also contributes to the first delay in maternal mortality, as mentioned by several studies [17].

### 4.2. Cultural Practices Related to Safe Motherhood

Traditional methods were extensively used to protect new mothers from health problems such as postpartum hemorrhage and colds. Some foods were forbidden for women who had given birth through Cesarean section or episiotomy because hot dish msakhen caused bleeding. Women in Morocco rely on traditional medicine to heal colds or bard nfass. Our findings are consistent with what Obermeyer has discovered in Morocco and other cultures over the last two decades [3,13,16].

Warmth is vital for the new mother. Only women who have given birth without an episiotomy benefit from a warm bath with massage and manipulation. Participants globally reported the same practice (kayjmaao laadam) [3].

The care and support of the family towards the new mother during the postpartum period are a ritual practice. A mother’s sanitary napkin is used in witchcraft to separate women from their husbands. In postpartum, new mother “nafssa” has a 40-day confinement and must not leave her bed for seven days. This is to prevent the “nfass bard” cold, bad eyesight, and sorcery. Fumigating the mother of the couple’s newborn and passing a clove of garlic around the room of the mother and her newborn, and putting it under her bed are also practices that help in fighting against bad odors that occur during this period. Our findings are supported by those of Raven and Obermeyer [3,12,13].

In general, mothers and mothers-in-law are the most influential people in recommending traditional practices such as eating “msakhen”, much hot food to warm the body, and recovering energy after giving birth and producing milk. These results are similar to those of Vietnamese women [18].

### 4.3. Cultural Practices Related to Newborn Health 

Fumigation-based chebba and lharmal are practiced every day before sunset. Successive crying without cause is a sign of the evil eye and sorra.

Some ritual practices could negatively affect the newborn’s health, such as applying olive oil and khol to dry and make the navel fall quickly. Comparable findings have been reported in Morocco since the 20th century [12,13,19]. Applying henna on the newborn’s body, which can distort the early detection of neonatal jaundice, is one of the most practiced rituals in the studied regions. Infant jaundice is a significant global cause of neonatal illness and death. It is a primary reason for hospitalization in the first week of birth worldwide, and one of the most prevalent causes of neurodevelopmental problems in developing nations [20,21].

## 5. Study Limitation

This study was carried out as a descriptive research design using a sample of 37 women drawn from three separate hospitals in three distinct regions of Morocco. As a result, the current study’s findings solely apply to women who took part in Morocco.

Because convenience samples were employed, the findings may not be generalizable. How the women learned about these behaviors or who proposed them was not addressed. However, because the participants’ age, educational level, employment, and number of children varied, a rather comprehensive picture of cultural postpartum beliefs and behaviors among Moroccan women in the provinces of Salé, Settat, and Essouira was acquired. More studies should be conducted on the education of healthcare personnel to enable them to give culturally appropriate care to mothers and newborns.

## 6. Conclusions

This research aids in raising the knowledge and understanding of maternal and newborn care in the Moroccan cultural setting. This new understanding has applications in instruction and research. The research also gives information to Moroccan groups that work to improve the health of mothers and children. This report promotes more research on this subject. It is critical to thoroughly understand the traditions and customs that regulate Moroccan families and women about motherhood, and how these traditions impact the lives of these women and their children.

Postpartum cultural beliefs and practices are common for Moroccan women and their babies. They are transmitted from mother to daughter from generation to generation. Many practices are beneficial to the health and wellbeing of mothers and babies, but some can be harmful. 

To reduce the power of these harmful traditional beliefs and practices regarding maternal health, the sensitization of women and the community at large to basic knowledge on the dangers associated with unskilled service uptake is critical, allowing for women to be aware of the consequences of harmful practices toward newborns and mothers, and adhere postnatal prenatal care. These requirements make it imperative that knowledge of cultural values be included in the training of midwives, nurses, and other professionals to increase their cultural awareness and skill opportunities for culturally appropriate care.

Midwives in particular and medical staff should know the cultural practices adopted by society, so they can deliver culturally appropriate messages to women, and convince them on how certain practices and beliefs can affect their health and the health of newborns. 

## Data Availability

Data are accessible upon reasonable request from the respective author.

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
