# Peer review of "Common Traditions, Practices, and Beliefs Related to Safe Motherhood and Newborn Health in Morocco"

_healthcare, 2023, doi:10.3390/healthcare11050769_

Round 1

Reviewer 1 Report

A very important article, not only for Morocco, but for many other countries where Moroccans immigrated to, like France and Israel.

There are some English mistakes in the following lines: 37 – "These" should be this. 95 – "present" should be presents. "lived" should be living. 136 – "beyond" should be before. 147 – "the risk is no longer" should be "there is no more risk". 159 – "chill" should be cold. 281 – "making henna" should be applying henna.

Author Response

Response to Reviewer 1 Comments

Point 1: A very important article, not only for Morocco but for many other countries where Moroccans immigrated to, like France and Israel.

Response 1: Thank you very much for agreeing with us on the intention of this manuscript.

Point 2: There are some English mistakes in the following lines: 37 – "These" should be this. 95 – "present" should be present. "lived" should be living. 136 – "beyond" should be before. 147 – "the risk is no longer" should be "there is no more risk". 159 – "chill" should be cold. 281 – "making henna" should be applying henna.

Response 2: Thank you very much for the reminder. We went through the entire manuscript to correct mistakes.

Reviewer 2 Report

The aim of this descriptive research was to determine the traditional practices related to maternal health in Morocco.

The title and abstract are appropriate for the content of the text, but the present study has not the form of scientific research. The results are presented in the form of description. There are also some fundamental concerns with the experimental design. The study was carried out as a descriptive research design using a sample of very few women. Generalized conclusions (drawn out the appropriate statistical analysis) cannot be presented due to the very small sample of women and the lack of standardization of investigates topics.

Moreover, I do not understand the aim of the research. It is commonly known that some practices (including cold postpartum treatment by traditional “medicine” and not seeking prenatal care after the first pregnancy experience) may negatively affect maternal health. It is obvious that women should be educated in this field, but I deem it unlikely that this subject should be analyzed in any scientific report. If the study design included the assessment of the health status of neonates in relation to their  mothers’ beliefs and practices (and optionally monitoring of this status for subsequent months of the early life),  it would sound much more scientifically.

This means the conclusions put forward by this manuscript are not warranted and I cannot approve the manuscript in this form.

Author Response

Response to Reviewer 2 Comments

Point 1: The aim of this descriptive research was to determine the traditional practices related to maternal health in Morocco.

The title and abstract are appropriate for the content of the text, but the present study has not the form of scientific research. The results are presented in the form of description. There are also some fundamental concerns with the experimental design. The study was carried out as a descriptive research design using a sample of very few women. Generalized conclusions (drawn out the appropriate statistical analysis) cannot be presented due to the very small sample of women and the lack of standardization of investigates topics.

Response 1:

Thanks for the comment. It turned out that some practices have persisted since the 1990s while others have only recently come to light. As you correctly noted, we worked with a very small sample size given the methodology we used because it is a qualitative study.

Unlike quantitative approaches which aim to establish statistical significance by sampling a predetermined number of subjects or elements, qualitative researchers do not usually begin a project with a predetermined sample size.

In qualitative research, there are no overall formal criteria for determining sample size and, therefore, no rules to suggest when a sample size is small or large enough for the study. Essentially, the ‘richness’ of data collected is far more important than the number of participants. This said, the researcher still requires insight into the size most likely to achieve the purpose, context, and richness of the data collected [V. Lopez and D. Whitehead, “Sampling data and data collection in qualitative research,” Nurs. Midwifery Res. Methods Apprais. Evid.-Based Pract., vol. 123, p. 140, 2013].

-Indeed, due to our sample’s small size, we cannot generalize our results. But we can conclude that this study has been able to draw realities when we can no longer draw them if we tackled a quantitative type study.

-This study supports the idea that customs, traditions, and beliefs continue to play a significant role in establishing social, gendered, cultural, and moral restrictions in Morocco.

Point 2:Moreover, I do not understand the aim of the research. It is commonly known that some practices (including cold postpartum treatment by traditional “medicine” and not seeking prenatal care after the first pregnancy experience) may negatively affect maternal health. It is obvious that women should be educated in this field, but I deem it unlikely that this subject should be analyzed in any scientific report. If the study design included the assessment of the health status of neonates in relation to their mothers’ beliefs and practices (and optionally monitoring of this status for subsequent months of early life),  it would sound much more scientifically.

This means the conclusions put forward by this manuscript are not warranted and I cannot approve the manuscript in this form.

Response 2: Thanks very much for your kind reminders.

-The theoretical framework of this study is based on an ontological starting point and understanding of the lifeworld in terms of how experiences are created from the behaviors, attitudes, practices, and perceptions of women and their families and communities.

-The task is to investigate some aspects of the lives of participants being studied, including finding out how these women view the situations they face, how they regard one another, and how they see themselves.

-To better address and do call on leaders to establish interventions to eliminate some traditions that may affect the health of mothers and newborns, this study aims to identify traditional practices related to postpartum and the health of newborns and to draw common practices that affect maternal and neonatal health among Moroccan women from different provinces studied. It also advocates for additional clinical research to credibly discuss the impact of using animal and plant matter to treat or prevent diseases that affect mothers and newborns.

-This study supports the idea that customs, traditions, and beliefs continue to play a significant role in establishing social, gendered, cultural, and moral restrictions in Morocco.

Reviewer 3 Report

The cultural context influences a women’s antenatal care and postpartum knowledge. This 10

study aims to determine the traditional practices related to maternal health in Morocco. 

There are several suggestions for the author:

1. In line 175 and 157, the superscript quotation mark is missing.

2. It is suggested to add the part of future improvement in the discussion.

Author Response

Response to Reviewer 3 Comments

Point 1: The cultural context influences a women’s antenatal care and postpartum knowledge. This 10 study aims to determine the traditional practices related to maternal health in Morocco.

Response1: Thank you very much for the reminder

Point 2:  In lines 175 and 157, the superscript quotation mark is missing.

Response2: Thank you so much, Revised accordingly

Point 3:  It is suggested to add the part about future improvement in the discussion.

Response 3: Thank you very much for agreeing with us on the intention of this manuscript. We have made revisions accordingly.

Reviewer 4 Report

Manuscript Number 2217428

Title: “Common traditions, practices, and beliefs related to safe motherhood and Newborn Health in Morocco”

The present study investigates the cultural aspects that may influence the well-being of mother and child in the very short term after birth. Three major themes emerged from the analysis: cultural practices related to pregnancy, cultural practices related to safe motherhood, and cultural practices described to newborn health. Participants were 37 women from three different Moroccan regions on the first postpartum day. Results showed that there are some practices that positively affect maternal health and they are the following: beliefs about pregnancy and postpartum, prolonged maternal rest, dietary precautions, and family support.

The limitations of this manuscript are theoretical. The introductory section is very poor and requires a much broader theoretical conceptualization.

Theoretical part

The impact of certain parental practices on child development during the child's socialization years has traditionally been studied in different cultural contexts. These are the importance of parental warmth and parental strictness, which have been studied in different models such as Baumrind's Y model and Maccoby and Martin's two-dimensional model. The authors should elaborate on these models and provide a clear explanation of the two main parental dimensions (warmth and strictness) and the parental styles that arise from the combination of these dimensions.

One of the firsts models about parenting was Baumrind´s Y model (Baumrind, 1968), that propose three parental styles: authoritative, authoritarian, and permissive; which corresponded to three modes of parental control (the same meaning that strictness), the authoritative control, the authoritarian control, and the lack of control (i.e., permissive control) (Baumrind, 1968).

Nevertheless, Maccoby and Martin's two-dimensional model (Maccoby & Martin, 1983) has had the greatest impact on parental socialization and has given rise to much empirical evidence on parental styles and child adjustment. This model states that parents use two independent parenting dimensions to socialize their children (i.e., warmth and strictness) (Climent-Galarza et al., 2022; Darling & Steinberg, 1993; Fuentes et al., 2022; F. Garcia & Gracia, 2009; Lamborn et al., 1991; Maccoby & Martin, 1983; Martinez et al., 2020; Martínez et al., 2021; Queiroz et al., 2020).

Parental warmth refers to the degree to parents show the children love, approval, acceptance and affection, give them their support, use dialogue with their children (Climent-Galarza et al., 2022), communication and reasoning with them (Martinez et al., 2020; Martínez et al., 2019), responsiveness, involvement, or implication (Darling & Steinberg, 1993; F. Garcia & Gracia, 2014; Martinez et al., 2020).

Parental strictness refers to the degree of parents use discipline towards their children, controlling and/or supervising their behavior, establishing norms for children’s behavior, and maintaining position of authority (Baumrind, 1991b; Darling & Steinberg, 1993) and parental

demands placed on children to promote compliance, i.e., the degree of imposition, authority, or rigidity (Climent-Galarza et al., 2022). Other labels used in the literature are demandingness, control, firmness (Darling & Steinberg, 1993; Steinberg, 2005), imposition (Martinez-Escudero et al., 2020) or supervision (O. F. Garcia et al., 2020).

According to Maccoby and Martin's two-dimensional model, four parenting styles emerges from the combination of the two main parenting dimensions (i.e., warmth and strictness): authoritarian (strictness but not warmth); authoritative (strictness and warmth), indulgent (warmth but not strictness) and neglectful (neither strictness nor warmth) (Climent-Galarza et al., 2022; Darling & Steinberg, 1993; Fuentes et al., 2022; O. F. Garcia et al., 2020; Maccoby & Martin, 1983; Perez-Gramaje et al., 2020; Queiroz et al., 2020; Villarejo et al., 2020).

This work is focused on studying factors that relate to the mother's well-being. Therefore, the discussion should add details about studies that focus on the well-being of the parents and not so much on the well-being of the child (Gomez-Ortiz & Sanchez-Sanchez, 2022).

This study also focuses on aspects related to the health of the child. For this reason, in discussion section other studies should be cited in which the relationship of certain practices or characteristics of parents with the well-being of their children has been seen. There is extensive literature focused on investigating which parenting style is associated with the best child adjustment.

Classical studies conducted in Anglo-Saxon contexts with European-American samples (mostly white middle-class families) state that the combination of parental warmth and parental strictness (i.e., the authoritative parenting) is associated with the best child psychosocial adjustment (Baumrind, 1991a; Darling & Steinberg, 1993; Lamborn et al., 1991; Steinberg et al., 1991; Steinberg et al., 1994). Nevertheless, other studies conducted in ethnic minority groups in the United States such as Chinese Americans (Chao, 2001) or African American (Deater-Deckard et al., 1996), and Arabs societies (Dwairy & Achoui, 2006) state that parental strictness without parental warmth (i.e., authoritarian parenting) is related to the best child adjustment in the short time.

The most recent evidence, conducted in European and Latin American countries, support the idea that parental warmth without strictness (i.e., indulgent parenting) is related to good child adjustment in the short (Fuentes et al., 2022; F. Garcia & Gracia, 2009; Martínez et al., 2019; Perez-Gramaje et al., 2020) and long term (Climent-Galarza et al., 2022; O. F. Garcia et al., 2018; O. F. Garcia & Serra, 2019; O. F. Garcia et al., 2020; Gimenez-Serrano et al., 2022; Martinez-Escudero et al., 2020; Villarejo et al., 2020).

Empirical part

In limitation section, it should be added the sample size. It is difficult to generalize results with such a small sample.

References

Baumrind, D. (1968, Authoritarian vs authoritative parental control. Adolescence, 3, 255-272.

Baumrind, D. (1991a). Effective parenting during the early adolescent transition. In P. A. Cowan, & E. M. Herington (Eds.), Advances in family research series. Family transitions (pp. 111-163). Lawrence Erlbaum Associates, Inc.

Baumrind, D. (1991b). Parenting styles and adolescent development. In R. M. Lerner, A. C. Petersen & J. Brooks-Gunn (Eds.), Encyclopedia of adolescence (pp. 746-758). Garland.

Chao, R. K. (2001). Extending research on the consequences of parenting style for Chinese Americans and European Americans. Child Development, 72, 1832-1843. https://doi.org/10.1111/1467-8624.00381

Climent-Galarza, S., Alcaide, M., Garcia, O. F., Chen, F., & Garcia, F. (2022). Parental socialization, delinquency during adolescence and adjustment in adolescents and adult children. Behavioral Sciences, 12(11)https://doi.org/10.3390/bs12110448

Darling, N., & Steinberg, L. (1993). Parenting style as context: An integrative model. Psychological Bulletin, 113(3), 487-496. https://doi.org/10.1037/0033-2909.113.3.487

Deater-Deckard, K., Dodge, K. A., Bates, J. E., & Pettit, G. S. (1996). Physical discipline among African American and European American mothers: Links to children's externalizing behaviors. Developmental Psychology, 32(6), 1065-1072. https://doi.org/10.1037/0012-1649.32.6.1065

Dwairy, M., & Achoui, M. (2006). Introduction to three cross-regional research studies on parenting styles, individuation, and mental health in Arab societies. Journal of Cross-Cultural Psychology, 37, 221-229. https://doi.org/10.1177/0022022106286921

Fuentes, M. C., Garcia, O. F., Alcaide, M., Garcia-Ros, R., & Garcia, F. (2022). Analyzing when parental warmth but without parental strictness leads to more adolescent empathy and self-concept: Evidence from Spanish homes. Frontiers in Psychology, 13https://doi.org/10.3389/fpsyg.2022.1060821

Garcia, F., & Gracia, E. (2014). The indulgent parenting style and developmental outcomes in South European and Latin American countries. In H. Selin (Ed.), Parenting Across Cultures (pp. 419-433). Springer. https://doi.org/10.1007/978-94-007-7503-9_31

Garcia, F., & Gracia, E. (2009). Is always authoritative the optimum parenting style? Evidence from Spanish families. Adolescence, 44(173), 101-131.

Garcia, O. F., Fuentes, M. C., Gracia, E., Serra, E., & Garcia, F. (2020). Parenting warmth and strictness across three generations: Parenting styles and psychosocial adjustment. International Journal of Environmental Research and Public Health, 17(20), 7487. https://doi.org/10.3390/ijerph17207487

Garcia, O. F., & Serra, E. (2019). Raising children with poor school performance: Parenting styles and short- and long-term consequences for adolescent and adult development. International Journal of Environmental Research and Public Health, 16(7), 1089. https://doi.org/10.3390/ijerph16071089

Garcia, O. F., Serra, E., Zacares, J. J., & Garcia, F. (2018). Parenting styles and short- and long-term socialization outcomes: A study among Spanish adolescents and older adults. Psychosocial Intervention, 27(3), 153-161. https://doi.org/10.5093/pi2018a21

Gimenez-Serrano, S., Garcia, F., & Garcia, O. F. (2022). Parenting styles and its relations with personal and social adjustment beyond adolescence: Is the current evidence enough? European Journal of Developmental Psychology, 19(5), 749-769. https://doi.org/10.1080/17405629.2021.1952863

Gomez-Ortiz, O., & Sanchez-Sanchez, C. (2022). Is the predisposition to have more children beneficial among parents with only one child? Evidence from Spanish parents? International Journal of Environmental Research and Public Health, 19(13), 7685. https://doi.org/10.3390/ijerph19137685

Lamborn, S. D., Mounts, N. S., Steinberg, L., & Dornbusch, S. M. (1991). Patterns of competence and adjustment among adolescents from authoritative, authoritarian, indulgent, and neglectful families. Child Development, 62(5), 1049-1065. https://doi.org/10.1111/j.1467-8624.1991.tb01588.x

Maccoby, E. E., & Martin, J. A. (1983). Socialization in the context of the family: Parent–child interaction. In P. H. Mussen (Ed.), Handbook of child psychology (pp. 1-101). Wiley.

Martínez, I., Murgui, S., Garcia, O. F., & Garcia, F. (2021). Parenting and adolescent adjustment: The mediational role of family self-esteem. Journal of Child and Family Studies, 30(5), 1184-1197. https://doi.org/10.1007/s10826-021-01937-z

Martínez, I., Murgui, S., Garcia, O. F., & Garcia, F. (2019). Parenting in the digital era: Protective and risk parenting styles for traditional bullying and cyberbullying victimization. Computers in Human Behavior, 90, 84-92. https://doi.org/10.1016/j.chb.2018.08.036

Martinez, I., Garcia, F., Veiga, F., Garcia, O. F., Rodrigues, Y., & Serra, E. (2020). Parenting styles, internalization of values and self-esteem: A cross-cultural study in Spain, Portugal and Brazil. International Journal of Environmental Research and Public Health, 17(7), 2370. https://doi.org/10.3390/ijerph17072370

Martinez-Escudero, J. A., Villarejo, S., Garcia, O. F., & Garcia, F. (2020). Parental socialization and its impact across the lifespan. Behavioral Sciences, 10(6), 101. https://doi.org/10.3390/bs10060101

Perez-Gramaje, A. F., Garcia, O. F., Reyes, M., Serra, E., & Garcia, F. (2020). Parenting styles and aggressive adolescents: Relationships with self-esteem and personal maladjustment. European Journal of Psychology Applied to Legal Context, 12(1), 1-10. https://doi.org/10.5093/ejpalc2020a1

Queiroz, P., Garcia, O. F., Garcia, F., Zacares, J. J., & Camino, C. (2020). Self and nature: Parental socialization, self-esteem, and environmental values in Spanish adolescents. International Journal of Environmental Research and Public Health, 17(10), 3732. https://doi.org/10.3390/ijerph17103732

Steinberg, L. (2005). Psychological control: Style or substance? In J. G. Smetana (Ed.), New directions for child and adolescent development: Changes in parental authority during adolescence (pp. 71-78). Jossey-Bass. https://doi.org/10.1002/cd.129

Steinberg, L., Lamborn, S. D., Darling, N., Mounts, N. S., & Dornbusch, S. M. (1994). Over-Time changes in adjustment and competence among adolescents from authoritative, authoritarian, indulgent, and neglectful families. Child Development, 65(3), 754-770. https://doi.org/10.1111/j.1467-8624.1994.tb00781.x

Steinberg, L., Mounts, N. S., Lamborn, S. D., & Dornbusch, S. M. (1991). Authoritative parenting and adolescent adjustment across varied ecological niches. Journal of Research on Adolescence, 1, 19-36.

Villarejo, S., Martinez-Escudero, J. A., & Garcia, O. F. (2020). Parenting styles and their contribution to children personal and social adjustment. Ansiedad y Estrés, 26(1), 1-8. https://doi.org/10.1016/j.anyes.2019.12.001

Author Response

Response to Reviewer 4 Comments

Point 1: The present study investigates the cultural aspects that may influence the well-being of mother and child in the very short term after birth. Three major themes emerged from the analysis: cultural practices related to pregnancy, cultural practices related to safe motherhood, and cultural practices described to newborn health. Participants were 37 women from three different Moroccan regions on the first postpartum day. Results showed that there are some practices that positively affect maternal health and they are the following: beliefs about pregnancy and postpartum, prolonged maternal rest, dietary precautions, and family support.

Response 1 : Thank you very much for pointing this out.

Point 2:The limitations of this manuscript are theoretical. The introductory section is very poor and requires a much broader theoretical conceptualization

Response 3: Your comment was taken into consideration, and the introduction section has been revised.

We grounded our work in a theoretical framework based on an ontological starting point and understanding of the lifeworld of the women as created from their behaviors, attitudes, practices, and perceptions and community and gender norms.[8], [9]To improve Morocco maternal and child health outcomes it is imperative to investigate how these women view their prenatal and postpartum situations [10].

The interaction between study participants and the researcher serves as the epistemological starting point for knowledge creation about their lived prenatal and postpartum experiences. It's possible that lifeworld knowledge is so ingrained in the culture that community members are unaware of the consequences of it for women’s and children’s health. Our study helps reveal this for populations in Morocco, [11] to improve outcomes for women, children and families and their healthcare team.  .

[8]           P. Aspers and S. Kohl, “Heidegger and socio-ontology: A sociological reading,” J. Class. Sociol., vol. 13, no. 4, pp. 487–508, Nov. 2013, doi: 10.1177/1468795X13480647.

[9]           “Heidegger M 2001 Sein und Zeit. Tübingen: Max Niemeyer Verlag.english version - Recherche Google.” https://www.google.com/search?q=Heidegger+M+2001+Sein+und+Zeit.+T%C3%BCbingen%3A+Max+Niemeyer+Verlag.english+version+&sxsrf=AJOqlzUYOjr5cSD76N62oh1donkSRu4jug%3A1676446735857&ei=D4zsY8O0M7SlkdUPtdyXmA0&ved=0ahUKEwiD-LLhgpf9AhW0UqQEHTXuBdMQ4dUDCA4&uact=5&oq=Heidegger+M+2001+Sein+und+Zeit.+T%C3%BCbingen%3A+Max+Niemeyer+Verlag.english+version+&gs_lcp=Cgxnd3Mtd2l6LXNlcnAQA0oECEEYAUoECEYYAFC9B1jAKGCTK2gBcAB4AIABbYgB3AuSAQQxMC42mAEAoAEBwAEB&sclient=gws-wiz-serp (accessed Feb. 15, 2023).

[10]         M. Hammersley and P. Atkinson, Ethnography: principles in practice, 3rd ed. London ; New York: Routledge, 2007.

[11]         D. F. Polit and C. T. Beck, Nursing research: generating and assessing evidence for nursing practice, Tenth edition. Philadelphia: Wolters Kluwer Health, 2017.

[12]         K. M. Borman, M. D. LeCompte, and J. P. Goetz, “Ethnographic and qualitative research design and why it doesn’t work,” Am. Behav. Sci., vol. 30, no. 1, pp. 42–57, 1986

Point 2:

Theoretical part

The impact of certain parental practices on child development during the child's socialization years has traditionally been studied in different cultural contexts. These are the importance of parental warmth and parental strictness, which have been studied in different models such as Baumrind's Y model and Maccoby and Martin's two-dimensional model. The authors should elaborate on these models and provide a clear explanation of the two main parental dimensions (warmth and strictness) and the parental styles that arise from the combination of these dimensions

One of the firsts models about parenting was Baumrind´s Y model (Baumrind, 1968), that propose three parental styles: authoritative, authoritarian, and permissive; which corresponded to three modes of parental control (the same meaning that strictness), the authoritative control, the authoritarian control, and the lack of control (i.e., permissive control) (Baumrind, 1968).

Nevertheless, Maccoby and Martin's two-dimensional model (Maccoby & Martin, 1983) has had the greatest impact on parental socialization and has given rise to much empirical evidence on parental styles and child adjustment. This model states that parents use two independent parenting dimensions to socialize their children (i.e., warmth and strictness) (Climent-Galarza et al., 2022; Darling & Steinberg, 1993; Fuentes et al., 2022; F. Garcia & Gracia, 2009; Lamborn et al., 1991; Maccoby & Martin, 1983; Martinez et al., 2020; Martínez et al., 2021; Queiroz et al., 2020).

Parental warmth refers to the degree to parents show the children love, approval, acceptance and affection, give them their support, use dialogue with their children (Climent-Galarza et al., 2022), communication and reasoning with them (Martinez et al., 2020; Martínez et al., 2019), responsiveness, involvement, or implication (Darling & Steinberg, 1993; F. Garcia & Gracia, 2014; Martinez et al., 2020).

Parental strictness refers to the degree of parents use discipline towards their children, controlling and/or supervising their behavior, establishing norms for children’s behavior, and maintaining position of authority (Baumrind, 1991b; Darling & Steinberg, 1993) and parental

demands placed on children to promote compliance, i.e., the degree of imposition, authority, or rigidity (Climent-Galarza et al., 2022). Other labels used in the literature are demandingness, control, firmness (Darling & Steinberg, 1993; Steinberg, 2005), imposition (Martinez-Escudero et al., 2020) or supervision (O. F. Garcia et al., 2020).

According to Maccoby and Martin's two-dimensional model, four parenting styles emerges from the combination of the two main parenting dimensions (i.e., warmth and strictness): authoritarian (strictness but not warmth); authoritative (strictness and warmth), indulgent (warmth but not strictness) and neglectful (neither strictness nor warmth) (Climent-Galarza et al., 2022; Darling & Steinberg, 1993; Fuentes et al., 2022; O. F. Garcia et al., 2020; Maccoby & Martin, 1983; Perez-Gramaje et al., 2020; Queiroz et al., 2020; Villarejo et al., 2020).

Response 2: Thank you very much for pointing this out.

The theoretical framework of this study is based on an ontological starting point and understanding of the lifeworld in terms of how experiences are created from the behaviors, attitudes, practices, and perceptions of women and their families and communities.

Indeed, this study supports the idea that customs, traditions, and beliefs continue to play a significant role in establishing social, gendered, cultural, and moral restrictions in Morocco.

So, these models can be used as a research framework if one is much more focused on studies that treat the relationship of parents with their children at an advanced age and as a study of psychosocial and or educational issues.

 The models you have proposed are very important and the way you will explain them is perfect,  but there are not relevant to our research objective.

Point 3 :

This work is focused on studying factors that relate to the mother's well-being. Therefore, the discussion should add details about studies that focus on the well-being of the parents and not so much on the well-being of the child (Gomez-Ortiz & Sanchez-Sanchez, 2022).

This study also focuses on aspects related to the health of the child. For this reason, in discussion section other studies should be cited in which the relationship of certain practices or characteristics of parents with the well-being of their children has been seen. There is extensive literature focused on investigating which parenting style is associated with the best child adjustment.

Classical studies conducted in Anglo-Saxon contexts with European-American samples (mostly white middle-class families) state that the combination of parental warmth and parental strictness (i.e., the authoritative parenting) is associated with the best child psychosocial adjustment (Baumrind, 1991a; Darling & Steinberg, 1993; Lamborn et al., 1991; Steinberg et al., 1991; Steinberg et al., 1994). Nevertheless, other studies conducted in ethnic minority groups in the United States such as Chinese Americans (Chao, 2001) or African American (Deater-Deckard et al., 1996), and Arabs societies (Dwairy & Achoui, 2006) state that parental strictness without parental warmth (i.e., authoritarian parenting) is related to the best child adjustment in the short time.

Response 3:

Thank you so much for your comment.

Our study is focused on knowledge about Moroccan women’s subjective perspectives, and cultural and traditional health beliefs that are
related to pregnancy and post-partum period.

this study supports the idea that customs, traditions, and beliefs continue to play a significant role in establishing social, gendered, cultural, and moral restrictions in Morocco.

We have included a number of studies that examine cultural customs, traditions, and beliefs concerning women in various contexts in the discussion section.

For the discussion section to be precise and understandable, we address each pertinent theme of our research separately.

Our study examines cultural practices and beliefs that have an impact on the health of pregnant women and newborns. In the discussion section, we cite works from contexts other than Morocco and identify commonalities in traditional practices that can have a negative impact on pregnant women's and newborns' health.

We draw attention to some cultural and traditional customs that are obscure in the society where these women reside.

Moroccan cultural traditions that support health can be both harmful and empowering for people.

Round 2

Reviewer 2 Report

Dear Authors,

The present study was carried out as a descriptive research design using a sample of very  few women. The results are presented only in the form of selected cited answers. Generalized conclusions cannot be presented due to the critically small sample of women and the lack of any standardization of investigates topics.

It is commonly known that some traditional "medicine" practices may negatively affect maternal health. It is obvious that women should be educated in this field, but I deem it unlikely that this subject should be analyzed in any scientific report. If the study design included the assessment of the health status of neonates in relation to their mothers’ beliefs and practices (and optionally monitoring of this status for subsequent months of early life),  it would sound much more scientifically.

This means the conclusions put forward by this manuscript are not warranted and I still cannot approve the manuscript in this form.

Author Response

Responses to Reviewer 2 Comment

Point 1:

The present study was carried out as a descriptive research design using a sample of very  few women. The results are presented only in the form of selected cited answers. Generalized conclusions cannot be presented due to the critically small sample of women and the lack of any standardization of investigates topics.

It is commonly known that some traditional "medicine" practices may negatively affect maternal health. It is obvious that women should be educated in this field, but I deem it unlikely that this subject should be analyzed in any scientific report. If the study design included the assessment of the health status of neonates in relation to their mothers’ beliefs and practices (and optionally monitoring of this status for subsequent months of early life),  it would sound much more scientifically.

This means the conclusions put forward by this manuscript are not warranted and I still cannot approve the manuscript in this form.

Response 1: Thank you very much for pointing this out.   This is a qualitative study conducted using the principles of this type of research. We designed and implemented our study based on current practices of this type of research design. We have referenced papers that support our methods and one of the authors (JTJr) is extensively published in very reputable journals using these methods.   Our results focus on elevating the voices and perspectives of Moroccan women about the cultural bases of their prenatal and postpartum practices and highlighting the themes that emerge from these women. This is the nature of qualitative studies. Our number of participants is actually higher than similar studies published in other journals. We followed the guidelines of qualitative research and recruited subjects until saturation of themes emerged.    We are well aware that this paper is not generalizable to other nations, or possibly to other communities in Morocco, but that does not make it unimportant. This type of work is very important for healthcare providers and health policy decision-makers in Morocco if they are going to be able to make more improvements in their pregnancy and infant health outcomes. Many of these individuals are men and women from every urban contexts that do not know about the traditions we highlight here. While we appreciate that cultural traditions are known to influence prenatal and postpartum practices, it is important to get this type of information gets out to readers in Morocco and other nations who might share similar practices so that healthcare providers can understand what might be going on in the home and community and family settings that influence maternal and infant health, and then develop appropriate intervention and communication strategies.    Furthermore, our paper is very important as it shows how we elevate the voices of women to understand their cultural context. Our approach in doing this is appreciated by maternal and child health researchers around the world and is important for nations in the Middle East North Africa (MENA) region, as we are the most public health poor region in the world. Elevating the voices of women in this area is very important for the advancement of true global health.    It is time that we start listening to the perspectives of women and integrate these perspectives into our decision-making. For too long, the voices and perspectives of women have not been included and inappropriate decisions about their care have been made. Our study is very important for the women of Morocco, as well as other women in the MENA region, as it shows them that we value their voices, perspectives, and cultural norms. The reviewer keeps asking for infant outcomes. While this would be wonderful, this is not possible in our context. The women and babies were discharged and went to their homes and villages and were not able to participate in follow-up as they must focus on the care of the family, the new baby, work duties, and community duties.    Thank you for the consideration of our response
